# A Non-Invasive Method of Estimating Populations of *Tomicus Piniperda* on Scots Pine (*Pinus Sylvestris* L.)

**DOI:** 10.3390/insects13111071

**Published:** 2022-11-20

**Authors:** Karol Zubek, Joanna Czerwik-Marcinkowska, Andrzej Borkowski

**Affiliations:** Department of Environmental Biology, Institute of Biology, Jan Kochanowski University, Uniwersytecka 7 Str., 25-406 Kielce, Poland

**Keywords:** larger pine shoot beetle, statistical model, population density estimation, biodiversity, forest management

## Abstract

**Simple Summary:**

Larger pine shoot beetle (*Tomicus piniperda*), which occupies a dominant position among secondary pests of Eurasian pine stands, is also a species whose activity leads to a number of favourable effects in relation to the functioning of ecosystems and broadly defined biodiversity. Thus it is necessary to have available an accurate, statistically based method for estimating its population. A fully non-invasive method for determining the numbers of *T. piniperda* can explain approximately 93% of the variation in the number of galleries in natural traps. The method may serve as a valuable supplement to existing methods used in the monitoring of *T. pinierda* populations. It may be used in nature reserves and in conservation-oriented forestry.

**Abstract:**

The fully non-invasive method presented here can be used to evaluate *Tomicus piniperda* L. population sizes in areas subject to strict protection. Data were collected in 2021–2022 in forests containing *P. sylvestris*, with different stand structures, in the Suchedniowsko-Oblęgorski Landscape Park, Poland. Entomological analyses were carried out on natural traps made from live uncolonised trees. Stepwise regression was used to describe the size of *T. piniperda* populations. From a set of features representing stem colonisation parameters, stem traits and habitat, the following had a significant effect (*p* < 0.05) on the total number of galleries of *T. piniperda* on stems: (1) the number of *T. piniperda* maternal tunnels in the sixth stem section covering 2.5% of the total length, (2) the length of the stem section with bark thickness greater than 7 mm, and (3) stand structure (homogeneous Scots pine stands). The model can explain 93% $\left(R_{adj}^{2}=0.9333\right)$ of the variability in the total number of *T. piniperda* galleries on trap trees. The mean relative error of estimation is 20.1%. The proposed method is particularly valuable in a climate context. The data obtained enable the prediction of the direct effects of climate change on the population dynamics of *T. piniperda* in natural forests.

## 1. Introduction

Climate change is expected to have a significant impact on the frequency and severity of disturbances to forest ecosystems [1]. Its consequences for biological systems are best studied using model organisms in near-natural and natural forests. Bark beetles are an important model group [2] and an important factor in conifer mortality [3,4]. A key species in forest ecosystems containing Scots pine (*Pinus sylvestris* L.) is *Tomicus piniperda* L. This is one of the best-described forest insects [5], and knowledge about the species is constantly expanding [6,7]. A biological sketch of this species indicates that (1) its biology and ecology are very well understood, (2) it is strongly expansive on new sites, (3) it is a cosmopolitan species on large and small scales, (4) it exhibits unique life traits, and (5) it is of high economic significance [2]. These characteristics make *T. piniperda* suitable for use as a model organism. The main limitation of the wide use of this species in protected areas is the invasiveness (whether to a lesser or greater degree) of the methods used for its monitoring. To evaluate the density of infestation, for example, on breeding material, it is necessary to cut the roots and branches from the stems and remove some or all of their bark [7,8,9]. The non-invasive methods currently used to estimate populations of the species suffer from low accuracy. Monitoring is usually carried out by visual surveying, for example, the identification of infested standing trees [10]. These qualitative methods produce results expressed in percentage terms, for example, and they do not provide any information about population numbers. The role of the pheromone information system is less important [11,12,13]. It is generally believed that the primary attraction phase plays a significant role in the colonisation of trees by *T. piniperda*. This phase consists of the beetles locating dead trees over a great distance through the emission of specific volatile compounds by the trees (mainly a-pinene) [5]. In the second half of the 20th century, the number of fallen shoots damaged by pine shoot beetles was used to calculate the population densities of insects of the genus *Tomicus* (Latreille) [14,15,16]. This method is now of less significance, as it enables only the calculation of an index related to density, at best proportional to it.

*Tomicus piniperda* has a unique characteristic that enables the development of a fully non-invasive method to evaluate its population numbers. Adults of *T. piniperda* reveal their presence by the typical boring dust containing brown bark and white wood grain (unique for this species) visible in bark crevices adjoining the entrance holes [2]. As the female bore the maternal tunnel, the male removed the boring dust from the surface of the stem. On the upper part of stems (for example, on trap logs, trap trees or windfalls), boring dust is very well visible in the form of sawdust points. It is not possible to count sawdust points on the underside of stems because the boring dust spills onto the ground. Evaluation of the total density of infestation of stems based on sawdust points counted only on the upper part of the stem is subject to two errors. The first results from the non-uniform colonisation of stems by the beetles. In the case of bark beetles, high variability is observed in the colonisation of individual stems or parts thereof [17]. The second error, known as the ‘edge error’, arises when counting sawdust points located on the boundary between the upper and lower parts of the stem. This error increases with the length of the stems analysed. One possible solution to these methodological problems is the use of models that enable the estimation of population size.

The aim of this study was to develop a new and fully non-invasive method, based on statistical techniques, to evaluate the size of *T. piniperda* populations on tree stems. In the proposed method, given the following data:i.the number of *T. piniperda* sawdust points on the sixth stem section covering 2.5% of the total length;ii.the length of the stem section with bark thickness greater than 7 mm;iii.the stand structure.

It is possible to calculate the total number of maternal tunnels of *T. piniperda* on stems, as well as the mean relative error of estimation.

## 2. Materials and Methods

### 2.1. Study Area

The study was conducted in the Suchedniowsko-Oblęgorski Landscape Park in central Poland (20°45′ E, 50°55′ N; 200–400 m above sea level) (Figure 1A). 50°55′ N; 200–400 m above sea level) (Figure 1A).

The Landscape Park contains a large forest complex (approximately 20,000 ha) comprising near-natural forests. The climatic conditions are characteristic of a mountain climate. Meteorological data for the period 2021–2022 were obtained from the Forest Data Bank [18]. The annual temperature amplitude in the valleys can exceed 60 °C. The annual mean temperature was 7.5 °C, and the annual mean precipitation was 650 mm. The growing season (the number of days with daily mean temperature above 5 °C) lasts from 1–5 April to 24–30 October. The prevailing winds in the area are south-westerly and westerly.

In July 2021, mixed and homogeneous Scots pine stands meeting the following criteria were selected for study: (1) age of pines over 80 years, and (2) proportion of *P. sylvestris* greater than 50% on a section measuring at least 2000 m. The mixed forest was found in the area of the Landscape Park in the forest sub-districts of Kruk and Wilczy Bór. Selected stands were located in Moist mixed coniferous forest (MMCF), Moist mixed broadleaved forest (MMBF) and Fresh mixed broadleaved forest (FMBF). The most important forest species are *Pinus sylvestris* L.—(Ps), *Picea abies* L. Karst.—(Pa), *Abies alba* Mill.—(Aa), *Fagus sylvatica* L.—(Fs) and *Quercus robur* L.—(Qr). Homogeneous Scots pine stands were not present in the area of the Landscape Park. Monocultures adjacent to the Park boundary in the forest sub-districts of Występa and Rejów were selected for the study (Figure 1B). Selected stands were located in Fresh coniferous forest (FCF) and Fresh mixed coniferous forest (FMCF). The most important forest species is *P. sylvestris*. In selecting stands for the study, it was assumed that bark beetle populations might be larger in homogeneous Scots pine stands than in stands with greater biodiversity.

Four distance zones were measured out every 400 m in the selected stands. In the Rejów and Występa sub-districts, the zones were measured directly from the edge of the stand: 1—edge zone, 2—400 m, 3—800 m, 4—1200 m. In the Kruk and Wilczy Bór sub-districts, the zones were measured starting 400 m from the edge of the stand: 1–400 m, 2–800 m, 3–1200 m, and 4–1600 m (Figure 1C). The following assumptions were adopted in determining the zones: (1) populations of *T. piniperda* may be higher in the edge zone of stands [7], and (2) the distance between zones allows detection of a potential source of *T. piniperda* reproduction [19]. A circular area of diameter 30 m was then established in each zone and assigned an appropriate label. Sample plots were marked out in the stands and coded with letters and numbers: a capital letter indicating the location of the plot (K—Kruk; WB—Wilczy Bór; R—Rejów; W—Występa) and a digit (1, 2, 3, 4) as the identifier of the sample plot. A full list of codes and the general characteristics of the sample plots are given in Table 1.

### 2.2. Fallen Shoots

On each sample plot, all pine trees were assigned numbers. Using the technique of simple random sampling with replacement and a random number generator [20], numbers were drawn to determine the numbers of 15 sample trees. Square plots of side 2 m were then marked out around the sample trees. Fallen shoots were collected from the plots in September and November 2021 and in spring 2022 after the snow had thawed. The density of fallen shoots was then determined for each sample plot.

### 2.3. Analyses of Trap Trees

In July 2021, two trees were selected on each sample plot—the healthy pines with the largest breast height diameter (the thickest tree, t_tc_) and the smallest breast height diameter (the thinnest tree, t_tn_). With increasing stem diameter in the bark at the thicker end, the number of *T. piniperda* maternal tunnels increases [7]. An additional criterion was the absence of technical defects along the entire length of the stems. A total of 32 sample trees were selected. Stem diameter and bark thickness at breast height were then measured. To measure the thickness of the bark, a hole was drilled into the bark using a 2 cm diameter hole saw. After the bark, the depth of the hole was measured (to an accuracy of 0.01 mm) using an electronic depth gauge. This method of measuring bark thickness gives a more accurate result than the measurement of bark cores obtained using an increment borer, for example. In the latter case, the bark is often fragmented during drilling, making precise measurement impossible.

In late January and early February 2022, the sample trees were felled and were then placed on supports at a height enabling colonisation by insects on the entire perimeter of the stems (Figure 1D). The trees prepared in this way will be further referred to as *trap trees*. These are natural traps that are used for bark beetle monitoring [10]. The stems of the trap trees were then divided into stem units covering 2.5% of the total length (40 equal units per stem). The bark thickness was measured at the midpoint of each unit by the method described above. In total, the bark thickness of 1280 stem units was measured (32 trap trees × 40 units). On each trap tree, the following were measured:-the stem diameter in the bark at the thicker and thinner ends;-the total length of the stem;-the length of the stem in thick bark (grey colour) and thin bark (red colour) and the bark transition area (grey and red colours).

The parameters of the trap trees are given in Table 2.

Next, the upper and lower parts of the stems of the trap trees were permanently marked along their entire length, with each part being one-half of the circumference. As a result of the division, the surface of each stem was divided into 80 sections lengthwise (into units) and laterally (into upper and lower parts).

After the spring flight of *T. piniperda* until the end of April, at a frequency of every 7 days, the boring dust on the upper stem sections was permanently marked with paint. The boring dust spots on the bark of the stems *sawdust points* were counted separately on each upper stem section immediately before the removal of the bark from the trap trees. In the second half of May, the trap trees were stripped of branches and debarked. The stems were divided into 80 sections, as described above. In each section, the maternal tunnels of pine shoot beetles were counted.

For each upper section, the mean relative error (*RE*) was calculated:(1)$$RE_{k}=\left|\frac{N_{mt}-N_{sp}}{N_{mt}}\right|\times 100$$
where *N_mt_* is the number of *T. piniperda* maternal tunnels in section *k* of the *P. sylvestris* trap trees (*k* = 1, 2, …, 40; each section covers 2.5% of the total length), and *N_sp_* is the number of *T. piniperda* sawdust points.

The total density of infestation on each trap tree was calculated by (1) summing the number of pine shoot beetle maternal tunnels from all sections and (2) calculating the stem surface area using the formulae given by Borkowski and Skrzecz, [21].

Differences in the density of infestation of trap trees by *T. piniperda* were tested (1) for individual forest sub-districts (post hoc Tukey HSD test), (2) for individual distance zones (Wilcoxon test), and (3) for the upper and lower parts of trap trees (one-sample *t*-test). The Spearman rank correlation coefficient was used to evaluate the relationship between the trap tree diameter and the length of the stem in thick bark. Normality and homogeneity of the variances were checked using the Shapiro–Wilk and Levene tests [22]. These analyses were performed using the STATISTICA v 13.3 statistical software package from StatSoft Inc., Tulsa, OK, USA [23].

### 2.4. Procedure for Selecting Explanatory Variables for Construction of the T. piniperda Population Model

The set of potential explanatory variables consisted of the following:Parameters of infestation of the upper section of stems (the number of *T. piniperda* maternal tunnels in the *k*th section of a *P. sylvestris* trap tree for *k* = 1, 2, …, 40; each section covers 2.5% of the total length).Stem features:
-the stem diameter in the bark at the thicker end;-the diameter and bark thickness at breast height;-bark thickness measured at the midpoint of the *k*th 2.5% stem unit of the trap tree;-length of the stem in thick bark, in thin bark, and in the bark transition area.
Habitat features:
-stand structure (mixed and homogeneous Scots pine stands);-site forest type (coniferous forest, broadleaved forest).


A selection was made of the best of the above parameters, namely those that were most strongly correlated with the explained variable while being weakly correlated with the other explanatory variables. In view of a large number of potential explanatory variables, the method of stepwise regression (forward selection) was used in the construction of the model. This method iteratively creates the ‘best’ regression model. It involves the successive inclusion of explanatory variables, which most significantly describe the total number of *T. piniperda* maternal tunnels on the trap trees. Stages of selection of explanatory variables:

Step 1. For each possible explanatory variable *x*_1_, *x*_2_, …, *x_k_* a separate one-variable model is constructed in the form:*y* = *b*_0_ + *b*_1_*x_j_*   *j* = 1, 2, …, *k*(2)
where *y* is the explained variable (the total number of maternal tunnels in the whole *P. sylvestris* trap tree), *b*_0_*, b*_1_*, b*_2_, *b*_3_ are parameters of the model, and the *x_j_* are values of the explanatory variables.

Step 2. For each of the remaining *k*-1 variables, a two-variable model is constructed:*y* = *b*_0_ + *b*_1 × 1_ + *b*_2_*x_j_*   *j* = 2, 3, …, *k*(3)

Step 3. For each of the remaining *k*-2 variables, a three-variable model is constructed:*y* = *b*_0_ + *b*_1 × 1_ + *b*_2_*x*_2_ + *b*_3_*x_j_*   *j* = 3, 4, …, *k*
(4)

In further steps, we proceed similarly with all other potential explanatory variables. The collinearity of explanatory variables was checked using the Variance Inflation Factor (VIF) [24]. It was assumed that the VIF should not exceed 5 [25]. The VIF was calculated according to Equation (5):(5)$$VIF_{j}=\frac{1}{1-R_{j}^{2}}$$
where $R_{j}^{2}$ is the multiple correlation coefficient between the variable *x_j_* and the other explanatory variables included in the model.

Habitat features were included in the model as dummy variables. For the purposes of regression modelling, a zero–one code was adopted:site forest type: 0 (broadleaved forest), 1 (coniferous forest);stand structure: 0 (mixed Scots pine stands), 1 (homogeneous Scots pine stands).

Following computation, it was checked whether the assumptions of the least squares method held. The verification process involved testing the properties of the residuals of the regression model:-the Shapiro–Wilk test was used to check whether the residuals were normally distributed;-the homoscedasticity of the distribution of regression residuals was analysed using White’s test [26]—Equation (6):


*W* = *nR*^2^(6)


where *n* is the number of observations, and *R*^2^ is the coefficient of determination of the auxiliary regression expressed by the equation
(7)$$e_{}{}^{2}=b_{0}+b_{1}X_{1}+b_{2}X_{2}+b_{3}X_{1}{}^{2}+b_{4}X_{2}{}^{2}+b_{5}X_{1}\times X_{2}$$

The critical value of the test is χ^2^ = χ^2^*_α_*_,*p*_, where *α* is the level of significance and *p* is the number of variables in the auxiliary regression.

The critical area is given by the inequality in Equation (7):*nR*^2^ > χ^2^
*_p_*_−1, α_(8)

The values used to measure the accuracy of the constructed model were the root mean square error *RMSE* Equation (8) and the mean relative error of estimation *sw_k_* Equation (9) [27,28,29]:(9)$$RMSE=\sqrt{\frac{\sum_{i=1}^{n}{\left(Y_{i}-{\hat{Y}}_{i}\right)}^{2}}{n-k-1}}$$
where *Y_i_* is the actual observation of the explained variable, ${\hat{Y}}_{i}$ is the predicted value of the observation, *n* is the number of observations, and *k* is the number of explanatory variables.
(10)$$sw_{k}=\sqrt{\frac{1}{n_{k}-2}\overset{n_{k}}{\underset{t=1}{\sum }}{\left(N_{ts_{t}}-a_{0k}-a_{1k}nTp_{k_{t}}\right)}^{2}}\frac{1}{{\bar{N}}_{ts}}$$
where ${\bar{N}}_{ts}=\frac{1}{n_{k}}\overset{n_{k}}{\underset{t=1}{\sum }}N_{ts_{t}}$; $N_{ts_{t}}$ is the total quantity of stem infestation (number of maternal tunnels) in the whole *P. sylvestris* trap tree *t*, $nTp_{k_{t}}$ is the number of *T. piniperda* maternal tunnels in stem section *k* of the *P. sylvestris* trap tree *t* (*k* = 1, 2, …, 40; each section is 2.5% of the total length), ${\bar{N}}_{ts}$ is the mean total number of trap trees, and *n_k_* is the number of trap trees for which a section *k* is present.

## 3. Results

### 3.1. Analysis of the Infestation of Trap Trees by T. piniperda

The mean quantity of fallen shoots on the sample plots was less than 0.2 shoots/m^2^. The quantity was higher only on sample plot W4, where it was approximately 0.6 shoots/m^2^. *Tomicus piniperda* had colonised all of the trap trees. That species co-occurred mainly with *Tomicus minor* Hartwig. The mean density of infestation of trap trees by *T. piniperda* in the Występa and Rejów forest sub-districts was higher than in the other sub-districts (ANOVA: *F*_3.28_ = 8.175, *p* = 0.0005; post hoc Tukey HSD test).

The colonisation of trap trees by *T. piniperda* in individual forest sub-districts is non-uniform (Figure 2).

The degree of colonisation of the thickest and thinnest stems in particular distance zones differed significantly (Wilcoxon test; *Z* = 3.31, *p* = 0.0009). The proportion of thick bark with increasing stem diameter (Spearman’s rank correlation; *r_s_* = 0.5307, *p* < 0.05).

*Tomicus piniperda* colonised mainly the thicker part of the stems, up to about 60% of their length (Figure 3).

The highest degree of colonisation (14.97 egg galleries per unit) was shown for the first 2.5% unit. The distribution of maternal tunnels on the trap trees showed a decrease in the density of *T. piniperda* with increasing distance from the thicker end of the stem. Colonisation by *T. piniperda* did not differ significantly between the upper and lower parts of the stems (one-sample *t*-test; *p* = 0.5631).

The relative error of estimation of maternal tunnels for stem sections with increasing distance from the thicker end of the stem. For sections 1–5, 17.4%, 18.9%, 14.2%, 9.5% and 5.3%. For section 6 and subsequent sections, less than 3%.

### 3.2. Model for Estimating T. piniperda Populations on Trap Trees

In the first step, the explanatory variable giving the highest value of the *F* statistic was introduced into the equation. The variable most strongly correlated with the explained variable (total number of maternal tunnels in whole *P. sylvestris* trap trees, *N_t_*) was the number of *T. piniperda* maternal tunnels in the sixth 2.5% stem section, *N*_6_ (Table 3).

The following equation was obtained Equation (10):
(11)$$N_{t}=20.912+25.094\times N_{6}$$

The number of maternal tunnels on the sixth stem section, used as an independent explanatory variable, explains approximately 86% of the variation in the colonisation of stems by *T. piniperda* (Table 3). A weakness of the model obtained is the insignificant value of the parameter *b*_0_ in the equation.

In the second step, the length of the stem section with bark thickness greater than 7 mm, *t_b_*, was introduced into Equation (11):(12)$$N_{t}=-67.466+22.690\times N_{6}+19.480\times t_{b}$$

When the second variable is included in the model, the proportion of the variance explained is above 92%. The root mean square error is 56.85, and the mean relative error of estimation is 21.7%. The parameters of the equation are significant.

In the third step, the variable representing stand structure, *s_s_* (homogeneous Scots pine stands), was introduced into Equation (12):(13)$$N_{t}=-94.137+20.941\times N_{6}+22.601\times t_{b}+51.315\times s_{s}$$

When the third variable is included in the equation, the proportion of the variance explained is above 93%. The root mean square error (52.75) and the mean relative error of estimation (20.1%) were lower than previously (Table 3). The parameters of the equation are significant.

In the next steps, the variables included were significant, but collinearity was detected. The calculated VIF exceeded the accepted cut-off value of 5.

In the final model, represented by Equation (12), the positive sign of the regression coefficients indicates that the total number of maternal tunnels in the whole of the *P. sylvestris* trap trees, *N_t_*, increases with an increase in (1) the number of *T. piniperda* maternal tunnels in the sixth 2.5% stem section, *N*_6_, and (2) the length of the section of the stem with bark thickness greater than 7 mm, *t_b_*. In addition, the colonisation of trap trees by *T. piniperda* is greater in homogeneous Scots pine stands than in mixed stands.

## 4. Discussion

### 4.1. Baseline Level Pine Shoot Beetles W Stands

The spatial distribution of the colonisation of trap trees (Figure 2) and fallen shoots does not exhibit a gradient as observed in stands growing close to centres of pine shoot beetle reproduction, such as timber yards [7,15]. This indicates that the local populations are not being augmented by individuals from brood material stored outside the forest area.

Pine shoot beetles are present in every stand containing Scots pine [2]. A key problem is the establishment of a baseline level for the population of pine shoot beetles in stands. In Sweden, in well-managed stands, the baseline level of the indicator of the population is less than 0.1 shoots/m^2^ [15,30], while in France, it ranges from 0.2 to 0.4 shoots/m^2^ [31]. In Poland, the baseline level determined on the basis of fallen shoots in a control plot (located 2000 m from a timber yard) is approximately 0.7 shoots/m^2^ [16]. As indicated by studies using labelled specimens of *T. piniperda*, this value may be. Most beetles migrate from a timber yard into surrounding stands to a distance of about 400 m, but the maximum dispersal can be up to 2000 m [19,32]. Pine stands within a distance of 500 m were consistently damaged during the whole of the growth period [16,33]. Potentially, given the absence of suitable shoots for feeding and the very large population of pine shoot beetles, the crowns of the pines in the control plot may have been damaged by beetles migrating from the timber yard. It may be assumed that the number of fallen shoots found in most of the sample plots, being less than 0.2 shoots/m^2^, represents a baseline level of population. The higher value in plot W4 (0.6 shoots/m^2^) was probably due to the presence of suitable host material remaining after the final felling in winter 2021. Based on the degree of colonisation of trap trees by *T. piniperda*, it may be assumed that the baseline level evaluated on the basis of fallen shoots may be even lower. The density of infestation of stems is lower in stands with greater species diversity. The above facts indicate that in well-managed stands and those with higher biodiversity, the baseline level may be: (1) below 0.1 shoots/m^2^ and (2) below 20 galleries/m^2^ on stems.

Fallen shoots may offer an original and easy way to sample *Tomicus* populations. This method is fully non-invasive but enables only the calculation of an index related to density. To make an accurate assessment of the pine shoot beetle population, it is necessary to establish the proportion between fallen shoots and the number of beetle-damaged shoots in the crowns. Studies have shown that beetles feed mainly in the top parts of shoots that do not fall to the ground [33,34,35,36,37]. Secondly, it is necessary to determine the number of shoots damaged by individuals in the population. Beetles can damage from one to as many as six shoots in a growing season [34,35,37,38,39,40].

### 4.2. A Population Size Model for T. piniperda

A key problem to be solved in the development of the model is the edge error. This arises when counting maternal tunnels and sawdust points at the boundary between units and between the upper and lower parts of the stem. The position of both of these signs of colonisation can be determined more easily and with greater precision after the bark has been removed from the stem. For this reason, maternal tunnels were taken as the basis for evaluating the mean relative error (1) and were used in the parameterisation of the model (10). Sawdust points in the thinner bark were clearly visible and separated. For the sixth and subsequent sections, the mean relative error is very small (less than 3%). This means that it is possible to use this parameter in the developed model. On trap trees, 20 m in length, the sixth 2.5% stem section is a section of length 50 cm located 2.5–3.0 m from the thicker end of the stem. An advantage of the model based on a 2.5% stem section, as opposed to a 5% or 10% section, is that the edge error is limited. This error increases as the length of the section increases. A second advantage of the model results from the position of the sixth section on the stem. As the distance from the thicker end of the stem increases, the thickness of the bark decreases. This facilitates the precise counting of sawdust points. In the thickest part of the stem, these points are less visible, being hidden by the thick, corky bark. This is confirmed by the higher mean relative error values in the thicker part of the stems. The study showed no differences in the colonisation of the upper and lower parts of the stems. However, during the counting of maternal tunnels, it was observed that the beetles had a preference for the side part of the stem. For this reason, in field studies, the borders of the sections must be very precisely marked.

The second variable included in the model is the length of the section of the stem with bark thickness greater than 7 mm. It is generally known that *T. piniperda* occurs mainly in thicker bark [41,42,43]. This is confirmed by the results of the present study. The proportion of thick bark increases with increasing stem diameter, and therefore the level of colonisation was greater on thicker trees than on thinner ones. These relationships have been observed on natural traps [7,17,44,45] and on wind-damaged trees [43].

The final variable included in the model represents habitat features. In pine monocultures, the density of infestation of trap trees is approximately twice as high as in stands with greater species diversity. Homogeneous Scots pine stands are more vulnerable to pest outbreaks, fires, or wind damage. The latter, in particular, is an important source of brood material for bark beetles. Pine windfalls constitute suitable breeding material for *T. piniperda*, because they cannot produce a defensive reaction, and also they secrete substances that trigger a primary attraction effect. They are colonised during the first and (mainly in the case of windfalls) the second growing season [6,43,46,47,48,49].

As the method is fully non-invasive and is not labour-intensive, *T. piniperda* monitoring can be carried out on the entire population of windfalls in the studied stand. For large samples (*N* > 30), the arithmetic mean is a good measure of the central tendency of a statistical series. Estimation of the degree of colonisation of pine stems by *T. piniperda* requires the following steps:

Stage 1. Evaluation of sawdust points on the sixth section;

division of trap trees into stem units covering 2.5% of the total length, starting from the thicker end. The sixth stem unit is then marked out. For a 30 m long trap tree, this is a unit of length 75 cm located 3.75–4.5 m from the thicker end of the stem;division of the perimeter of the sixth unit into two equal parts (upper and lower sections);counting of sawdust points on the upper section.

Stage 2. Measurement of the length of the stem section with bark thickness greater than 7 mm;

identification of a unit with bark thickness greater than 7 mm;measurement of the length of the part of the stem with bark thickness greater than 7 mm. The measurement covers the section from the thicker end of the stem to the further end of the unit, determined in step 1.

Stage 3. Evaluation of stand structure (homogeneous Scots pine stands, mixed stands).

This is an indirect method of evaluating the size of the *T. piniperda* population in a surveyed stand, as we assume that the colonisation of stems by this species is directly proportional to its population size.

## 5. Conclusions

*Tomicus piniperda* has typical characteristics of a model organism, and it is, therefore, important to have a non-invasive, accurate, statistically based method of assessing its population size. This will allow it to be used as a model species in the most valuable natural areas, subject to strict protection. Model species are particularly valuable in a climatic context. The data obtained enable prediction of the direct effects of climate change on population dynamics in near-natural and natural forests.

This study has identified variables determining the colonisation of trap trees by *Tomicus piniperda*. A linear model describing population size was obtained with three statistically significant explanatory variables. The number of *T. piniperda* maternal tunnels on the sixth stem section covers 2.5% of the total length, the length of the stem section with bark thickness greater than 7 mm, and the stand structure have a positive effect on the colonisation of stems by *T. piniperda*. The low mean relative error calculated for sawdust points indicates that they can be used in the model as a parameter for colonisation.

The method of assessing *T. piniperda* population size based on the counting of sawdust points is fully non-invasive. It does not require tree felling, branch trimming, or debarking of stems. The low labour intensity of the method means that it is possible to evaluate *T. piniperda* populations on a large sample. These two elements enable the monitoring of this species on the entire population of natural traps in the study area. This is important during the validation and modification of the method according to habitat and standing conditions. It is desirable to develop global models applicable to larger areas and local models for use in specific stands.

An advantage of the model’s design is that it does not require the use of advanced statistical procedures. Because the method does not require a large amount of time for data preparation, the population dynamics of *T. piniperda* can be tracked continuously in the field. This is of great practical importance, as it means that the cause of an increase in population density can be identified at a given place and time. This allows additional surveys and observations to be carried out in parallel with the ongoing population evaluation.

In national parks and nature reserves, the method can be a valuable complement to existing methods for monitoring *T. piniperda*. In forestry practice, the developed method, in combination with the method of visual estimation, can be used when delimiting areas at different levels of risk from bark beetles.

## Figures and Tables

**Figure 1 insects-13-01071-f001:**
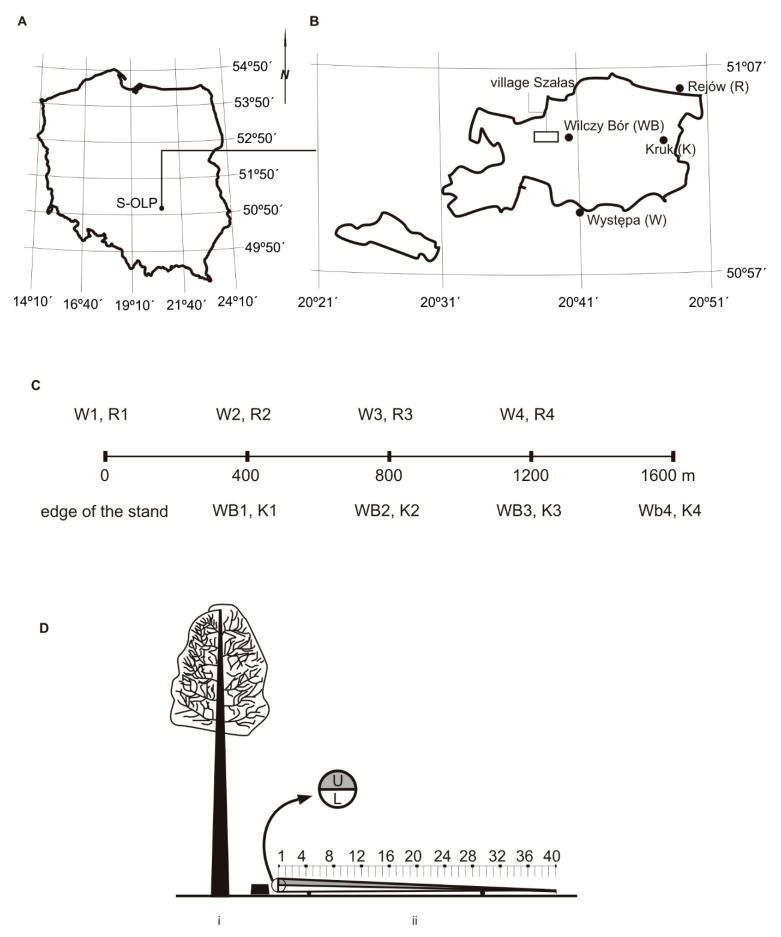
Study region. (**A**) Grid of the SINUS (System of Information on Natural Environment) system for Poland; S-OLP—Suchedniowsko-Oblęgorski Landscape Park (**B**) location of Forest sub-district (**C**) location of sample plots in respective distance zones; W1-4, R1-4, WB1-4 and K1-4 for plot codes—see Table 1 (**D**) Healthy pine (i) used as a trap tree (ii); letters U and L indicate the upper and lower section of the trap tree, respectively. Numbers 1, 2, …, and 40 indicate stem units.

**Figure 2 insects-13-01071-f002:**
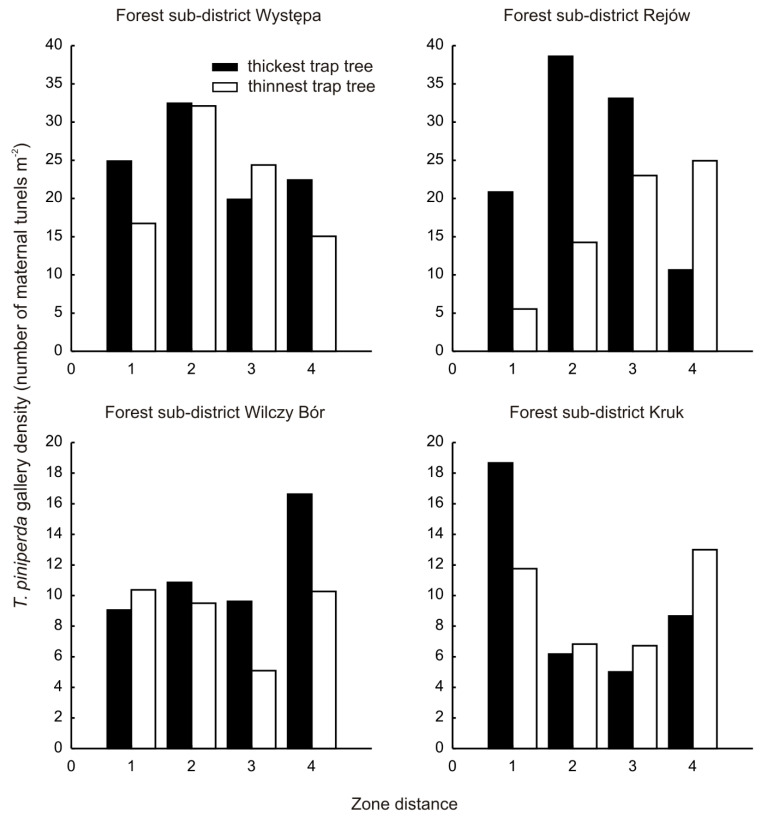
Colonisation of trap trees by *T. piniperda* in respectively Forest sub-districts.

**Figure 3 insects-13-01071-f003:**
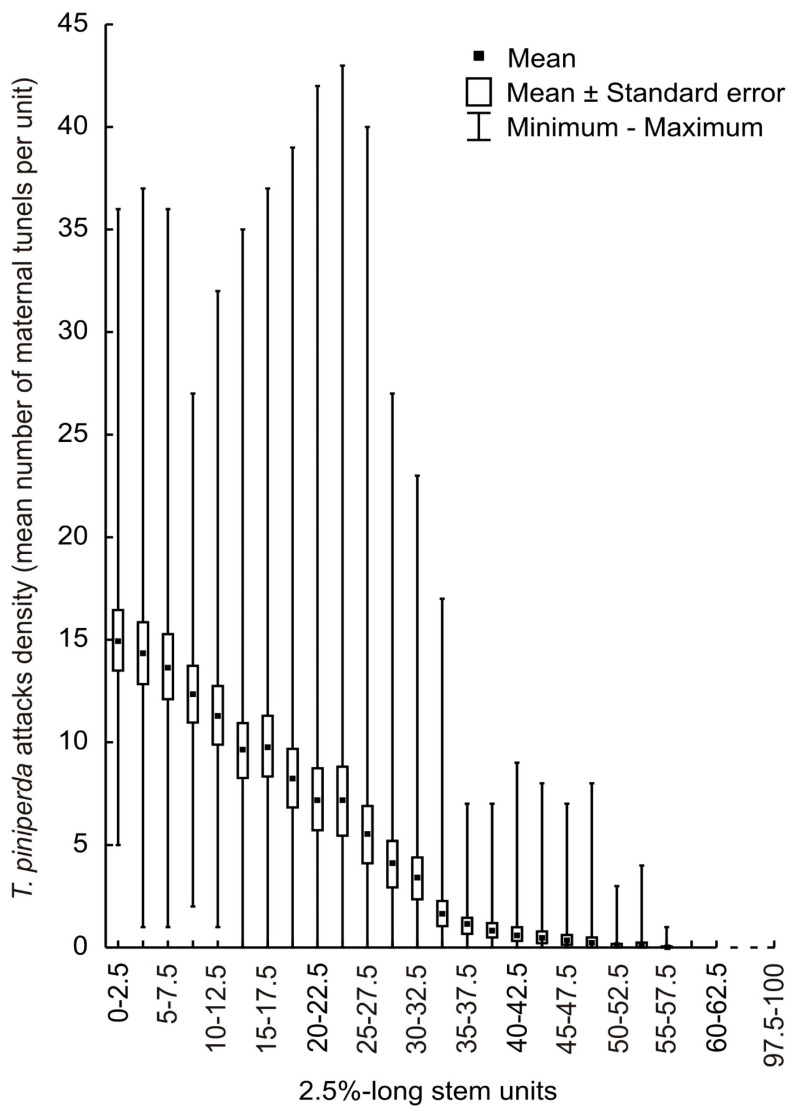
Distribution of *T. piniperda* on *P. sylvestris* trap trees in 2.5%-long stem units in sample plots.

**Table 1 insects-13-01071-t001:** Locality and stand data.

Forest Sub-District	Sample Plot No ^1^	Distance from the Edge of the Stand (m)	Forest Type Site ^2^	Age of Pine (Years) ^3^	Structures of Forest Stands (Tree Species in 10 ths) ^4^	
					Single-Storied	Two-Storied
Występa	W1, 2	0, 400	FCF	98	Ps 10	
	W3, 4	800, 1200	FMCF	96	Ps 10	
Rejów	R1	0	FMCF	85	Ps 10	
	R2	400	FMCF	85	Ps 10	
	R3, 4	800, 1200	FMCF	85	Ps 10	
Wilczy Bór	WB1, 2	400, 800	MMCF	89	Ps 9, Bp 1	Bp 4, Pa 2, Qr 2, Aa 2
	WB3	1200	MMBF	79	Ps 10	Pa 5, Qr 2, Bp 2, Aa 1
	WB4	1600	MMCF	84	Ps 10	Aa 5, Pa 3, Bp 1, Qr 1
Kruk	K1	400	FMBF	100	Ps 9, Qr 1	Qr 9, Aa 1
	K2	800	FMBF	110	Ps 7, Qr 2, Aa 1	Aa 6, Qr 4
	K3	1200	FMBF	105	Ps 6, Aa 2, Qr 1	Aa 9, Qr 1
	K4	1600	FMBF	95	Ps 5, Qr 3, Aa 2	Aa 9, Qr 1

^1,2,4^ See the “Materials and methods” section. ^3^ Estimated from a tree felled. On each sample plot, sample trees were designated for the evaluation of pine shoot beetle populations based on (1) fallen shoots—shoots damaged by beetles and fallen to the ground, and (2) maternal tunnels on trap trees.

**Table 2 insects-13-01071-t002:** Characteristics of Scots pine trap trees felled in the investigated stands.

Sample Plot ^1^	Trap Tree ^2^	Length of the Trunk (m)	Diameter Outsider Bark at Thicker End (cm)	Diameter at Breast Height (cm)	Thickness of the Dbh Bark (mm)	Length of the Bark Transition Area on the Trunk (m)
W1	t_tc_	22.0	31.75	27.00	19.01	5.7–7.7
	t_tn_	17.0	27.25	20.25	14.94	2.5–4.2
W2	t_tc_	26.2	42.75	39.75	32.98	6.6–13.0
	t_tn_	21.6	30.75	22.75	21.89	4.3–6.5
W3	t_tc_	29.0	55.5	43.00	26.41	11.0–17.0
	t_tn_	22.6	31.0	26.50	16.63	4.7–6.8
W4	t_tc_	26.0	33.5	28.50	19.71	6.0–8.4
	t_tn_	22.8	27.75	22.25	17.18	7.1–8.6
R1	t_tc_	23.2	30.75	25.00	22.80	3.2–4.7
	t_tn_	19.2	20.75	17.75	11.22	3.1–7.2
R2	t_tc_	29.6	51.75	45.25	21.31	8.1–13.2
	t_tn_	25.6	23.75	21.00	13.07	3.8–5.9
R3	t_tc_	28.0	52.00	45.50	28.37	8.9–12.1
	t_tn_	24.8	31.25	27.00	16.14	3.5–5.7
R4	t_tc_	27.2	35.25	30.25	18.01	3.3–4.4
	t_tn_	21.2	26.25	21.75	8.71	8.1–10.3
WB1	t_tc_	25.6	35.25	32.25	19.05	4.4–6.1
	t_tn_	22.0	26.25	20.75	10.97	3.5–5.3
WB2	t_tc_	27.6	48.50	41.75	21.97	8.5–12.0
	t_tn_	24.4	32.50	24.50	17.64	7.3–9.1
WB3	t_tc_	28.4	57.00	49.25	20.95	7.5–10.6
	t_tn_	22.8	30.25	25.50	17.12	8.0–10.0
WB4	t_tc_	24.8	33.75	30.75	19.34	6.5–7.9
	t_tn_	24.0	27.0	23.75	19.47	8.5–11.3
K1	t_tc_	30.4	55.75	50.50	40.72	9.1–13.1
	t_tn_	26.4	33.25	30.75	18.57	7.9–10.8
K2	t_tc_	29.5	47.0	40.25	28.51	8.3–12.1
	t_tn_	25.5	40.0	33.75	18.21	6.8–10.5
K3	t_tc_	31.2	67.0	53.00	20.88	8.0–12.0
	t_tn_	26.8	40.0	34.00	21.78	7.5–10.0
K4	t_tc_	27.6	60.0	47.25	38.97	6.4–8.6
	t_tn_	22.0	29.5	24.5	15.41	5.6–9.0

^1^ For plot codes, refer to the Materials and Methods section. ^2^ For trap tree codes, refer to the Materials and Methods section.

**Table 3 insects-13-01071-t003:** Parameters and basic statistics for Equations (8)–(10).

No. Equation	Name of Variable	Value of Parameter	SE	Value *t*-Statistics	Probability Level	VIF	*R*	*R* ^2^ * _adj_ *	RMSE	ANOVA		Mean Relative Error of Estimation (%)
										*F*-Value	*p*-Level	
8	Intercept	20.912	21.6419	0.9663	0.342		0.9321	0.8645	75.214	198.77	<0.001	28.7
	*N* _6_	25.094	1.7799	14.0985	<0.001							
9	Intercept	−67.466	24.4905	−2.7548	0.010							
	*N* _6_	22.690	1.4338	15.8250	<0.001	1.14	0.9631	0.9226	56.850	185.72	<0.001	21.7
	*t_b_*	19.480	4.0194	4.8489	<0.001							
10	Intercept	−94.137	25.3328	−3.7160	<0.001		0.9694	0.9333	52.754	145.68	<0.001	20.1
	*N* _6_	20.941	1.5195	13.7817	<0.001	1.48						
	*t_b_*	22.601	3.9518	5.7192	<0.001	1.28						
	*f_s_*	51.315	21.5349	2.3829	0.024	1.33						

*N*_6_—the number of *T. piniperda* egg galleries in the sixth stem section covering 2.5% of the total length; *t_b_*—the length of the stem section with bark thickness greater than 7 mm; *f_s_*—forest stand structure; SE—standard error; VIF—Variance Inflation Factor; *R*—coefficient of correlation; *R*^2^*_adj_*—adjusted *R*-squared. RMSE—root mean square error.

## Data Availability

All relevant data are within the paper.

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
