# Peer review of "A Non-Invasive Method of Estimating Populations of Tomicus Piniperda on Scots Pine (Pinus Sylvestris L.)"

_insects, 2022, doi:10.3390/insects13111071_

Round 1
Reviewer 1 Report
The manusript titled "A non-invasive method of estimating populations of Tomicus piniperda on Scots pine (Pinus sylvestris L.)" is well written and carefully designed. I find obtained result usefull for forestry practise, especially in protected areas. I recommend the manuscript for publication with only minor revision:
In my opinion Fig. 2 is redundant and just the brief description in text is enough.
line 323 - "in" instead of "w"
Author Response
First, we would like to thank for comments and suggestions for changes. In the revised version of the article, we corrected the fragments of the text that required correction (Figure 2 removed).
Reviewer 2 Report
Minor language and formatting suggestsion were made on the MS.
Table 1: no need for horizontal lines within the table. Tables and figures need to be self-udnerstandable, so do not write footnotes as you did under Table 1 - write the codes out.
Fig 2 is unnecessarily large - there are only 4 data points, reduce size considerably
Fig 3 needs to present variability data, too.
Your own results has to be presented in past tense, not present.
Fig 4 - you probalby do not need the hrozintal axis above 62.5

Author Response
First, we would like to thank for their detailed remarks and suggested changes. We am grateful to for drawing attention to evident errors and inaccuracies, which arose at the stage of translation of the manuscript into English.
In the corrected version of the paper, we have incorporated all of the suggestions, and we have amended passages of the text that required correction.
Below we provide detailed responses.
- Table 1: no need for horizontal lines within the table. Tables and figures need to be self-udnerstandable, so do not write footnotes as you did under Table 1 - write the codes out.
Re. 1. Horizontal lines have been deleted in Tables 1 and 2. The codes in Table 1 (forest type site, most important forest species ) are described in the material and methods section.
“Selected stands were located in Moist mixed coniferous forest (MMCF), Moist mixed broadleaved forest (MMBF) and Fresh mixed broadleaved forest (FMBF). The most important forest species are Pinus sylvestris L – (Ps), Picea abies L. Karst. – (Pa), Abies alba Mill. – (Aa), Fagus sylvatica L – (Fs) and Quercus robur L – (Qr). Homogeneous Scots pine stands were not present in the area of the Landscape Park. Monocultures adjacent to the Park boundary in the forest sub-districts of WystÄ™pa and Rejów were selected for the study (Figure 1B). Selected stands were located in Fresh coniferous forest (FCF) and Fresh mixed coniferous forest (FMCF). The most important forest species is P. sylvestris”.
- Fig 2 is unnecessarily large - there are only 4 data points, reduce size considerably.
Re. 2. As suggested by the Reviewer #1, Figure 2 has been removed
- Fig 3 needs to present variability data, too.
Re. 3. Figure 3 (now Figure 2) shows the infestation density of individual trap trees. Therefore, variability cannot be calculated.
- Your own results has to be presented in past tense, not present.
Re. 4. The text has been redrafted
- Fig 4 - you probalby do not need the hrozintal axis above 62.5
Re. 5. Figure 4 (now Figure 3) has been corrected.
In addition, language errors marked in the manuscript were corrected.
Reviewer 3 Report
The authors have designed and described new method of non-invasive way of population estimation of a bark beetle species. I believe that method is well constructed and performed and in general I do not have objectives. My main disagreement I have, is related to general assumption of the research – authors claim that using pheromone traps in protected areas is a “artificial interference” and so they promote own method instead. In my opinion pheromones against bark beetles in general, are : 1. Species specific, so it is environment friendly method; 2. Well studied and effective; 3. Used worldwide also in protected areas. Because of that I do not see a conflict in using pheromone traps, as this is the way that requires less labor and seems, to me, as more effective. I have noted quite a lot of editing errors, but I believe it is deal for editors (like for example Figure 1. line 92 – should be D- Healthy pine).
Author Response
First, we would like to thank for comments and suggestions for changes. In the revised version of the article, we corrected the fragments of the text that required correction.
- My main disagreement I have, is related to general assumption of the research – authors claim that using pheromone traps in protected areas is a “artificial interference” and so they promote own method instead ...... Because of that I do not see a conflict in using pheromone traps, as this is the way that requires less labor and seems, to me, as more effective.
Re. 1. As suggested by the Reviewer, the text "The installation of pheromone traps in areas subject to strict protection is considered a form of ‘artificial’ interference in the forest ecosystem" was changed to " The role of the pheromone information system is less important [11,12,13]. It is generally believed that the primary attraction phase plays a significant role in the colonization of trees by T. piniperda. This phase consists of the beetles locating dead trees over a great distance through the emission of specific volatile compounds by the trees (mainly a-pinene) [5]".